# Knowledge and Attitudes of Medical and Health Science Students in the United Arab Emirates toward Genomic Medicine and Pharmacogenomics: A Cross-Sectional Study

**DOI:** 10.3390/jpm10040191

**Published:** 2020-10-24

**Authors:** Azhar T. Rahma, Mahanna Elsheik, Iffat Elbarazi, Bassam R. Ali, George P. Patrinos, Maitha A. Kazim, Salma S. Alfalasi, Luai A. Ahmed, Fatma Al Maskari

**Affiliations:** 1Institute of Public Health, College of Medicine & Health Sciences, United Arab Emirates University, Al Ain 17666, UAE; 201280026@uaeu.ac.ae (A.T.R.); mahanna.s@uaeu.ac.ae (M.E.); ielbarazi@uaeu.ac.ae (I.E.); 201606441@uaeu.ac.ae (M.A.K.); 201607157@uaeu.ac.ae (S.S.A.); luai.ahmed@uaeu.ac.ae (L.A.A.); 2Zayed Center for Health Sciences, United Arab Emirates University, Al Ain 17666, UAE; bassam.ali@uaeu.ac.ae (B.R.A.); gpatrinos@upatras.gr (G.P.P.); 3Department of Pathology and Genomics and Genetics, College of Medicine and Health Sciences, United Arab Emirates University, Al Ain 17666, UAE; 4Department of Pharmacy, School of Health Sciences, University of Patras, 26504 Patras, Greece

**Keywords:** knowledge, attitudes, genomic medicine, pharmacogenomics, barriers, UAE students

## Abstract

Medical and health science students represent future health professionals, and their perceptions are essential to increasing awareness on genomic medicine and pharmacogenomics. Lack of education is one of the significant barriers that may affect health professional’s ability to interpret and communicate pharmacogenomics information and results to their clients. Our aim was to assess medical and health science students’ knowledge, attitudes and perception for a better genomic medicine and pharmacogenomics practice in the United Arab Emirates (UAE). A cross-sectional study was conducted using a validated questionnaire distributed electronically to students recruited using random and snowball sampling methods. A total of 510 students consented and completed the questionnaire between December 2018 and October 2019. The mean knowledge score (SD) for students was 5.4 (±2.7). There were significant differences in the levels of knowledge by the year of study of bachelor’s degree students, the completion status of training or education in pharmacogenomics (PGX) or pharmacogenetics and the completion of an internship or study abroad program (*p*-values < 0.05. The top two barriers that students identified in the implementation of genomic medicine and pharmacogenomics were lack of training or education (59.7%) and lack of clinical guidelines (58.7%). Concerns regarding confidentiality and discrimination were stated. The majority of medical and health science students had positive attitudes but only had a fair level of knowledge. Stakeholders in the UAE must strive to acquaint their students with up-to-date knowledge of genomic medicine and pharmacogenomics.

## 1. Introduction

Personalized medicine is the practice in which medical treatments are unique to the patient and chosen based on the patient’s genomic profile [1]. Genomic medicine and pharmacogenomics (PGX) are a central tool used in personalized medicine [2]. PGX is a relatively new field that studies and explores the effect of genetic variation on drug response [1,2,3]. The terms pharmacogenomics and pharmacogenetics are frequently used interchangeably, however, they do have different meanings. Pharmacogenetics observes the influence of single genes on a patient’s drug response, while PGX observes the influence of the entire genome on the drug response to therapy. [4] PGX aims to improve drug efficacy and minimize drug toxicity [1,2,3].

The completion of the Human Genome Project in 2003 propelled personalized medicine and made the concept more popular between clinicians, and it encouraged the implementation of genomics education [5,6,7]. This was displayed in a global survey conducted to evaluate the education progress of PGX, showing that 82.1% of the programs began the implementation of PGX topics after the completion of the Human Genome project [6]. In 2007, four years after the Human Genome Project was completed, the Food and Drug Administration (FDA) implemented pharmacogenetics, labeling changes to warfarin to indicate that genetic makeup can affect dosage requirements and risks [7,8]. This change, among several others, was deemed an important tangible step in PGX and personalized medicine [4,7,8].

Future advances in genomic medicine and PGX will require health professionals to be equipped with the knowledge and tools in order to fully apply and implement PGX in clinical practice as best as possible [9,10,11,12]. Despite the emphasis and evidence on the importance of genomic medicine and PGX in clinical practice, many healthcare professionals express a lack of confidence in the implementation of PGX in practice [7,13]. This is fairly attributed to lack of education, a widely highlighted barrier, which can lead to knowledge gaps and difficulty in interpreting and communicating PGX results [13]. Medical and health science students represent future health professionals, and their perceptions are essential to increasing awareness on personalized medicine and PGX [1,11,12,13]. Particularly, pharmacists, as drug experts, are considered integral in the clinical implementation of PGX due to the nature of their education and background [4,7,14,15]. In order to increase genomic medicine and PGX awareness and competency among medical and health science students, their knowledge, attitudes, and practice towards genomic medicine and PGX should be evaluated.

Little is known and studied regarding the genomic medicine and PGX educational environment and medical and health science students’ perceptions towards the practice in Middle Eastern and, more specifically, Gulf Cooperation Council (GCC) countries. Two studies have been conducted in Qatar and Kuwait. Synonymous with other studies, both Qatar and Kuwait respondents identified lack of knowledge to be one of the challenges, despite the positive attitudes in PGX clinical implications [16,17]. This study aims to assess medical and health science students’ knowledge and attitudes regarding genomic medicine and PGX to attempt to bridge this knowledge gap and gain insight on their opinions and views on the practice of personalized medicine and PGX.

## 2. Materials and Methods

We conducted a cross-sectional study using a validated and piloted questionnaire. The targeted sample included undergraduate and postgraduate medical and health science students (medicine, pharmacy, laboratory, medical imaging, radiology, radiography, biochemistry, biomedical sciences, dentistry, pharmacology, physiology, psychology, public health, and occupational health) in the United Arab Emirates (UAE), as they are the future adopters of genomic medicine and PGX. We employed random selection sampling techniques, in which we contacted all the universities and colleges in the UAE that offer degrees in medicine, pharmacy, laboratory, and nursing and asked them to distribute the questionnaire among their students. Moreover, we employed snowball sampling where existing students recruit future subjects from among their acquaintances that meet our inclusion criteria. The survey was administered electronically between December 2018 and October 2019.

The questionnaire was designed based on the literature to explore and identify knowledge, awareness, attitudes, behavior and interest in genomic medicine and PGX among medical and health science students. It encompassed validated questions used in the Public Understanding and Attitudes towards Genetics and Genomics (PUGGS) Questionnaire [18], the United States [19] and Southeast Europe [2]. We piloted the questionnaire among 50 medical and health science students and amended it accordingly. The questionnaire was administered in English and it was divided into 3 sections:

Section 1: Demographic data: age, gender, faculty, year of study, major, type of university—government or private.

Section 2: Knowledge: we asked nine questions about genomic medicine and PGX. Based on a literature review, a sum of more than 75% correct answers will indicate good knowledge in the field.

Section 3: Attitudes of the students’ ethical, social, and economic implications and their perceived barriers for the full implementation of genomic medicine and pharmacogenomics in the UAE.

We calculated the sample size using the formula for cross-sectional study (1.96^2^ × P (1−P) ÷ d^2^), where: P = 48 (48% is the prevalence of the knowledge of genomics among medical and health science students that was extracted from the literature of similar studies) and d = 0.05. The sample size (students) = 3.84 × 0.48 (1−0.48) ÷ 0.0025 = 383 students. Similar regional studies showed an average response rate of 84%, therefore an additional 61 students to reach a final sample size of 444 students.

We used International Business Machines Corporation Statistical Package for the Social Sciences (IBM SPSS) Statistics version 26 to analyze the data. Descriptive statistics (means (standard deviation, SD) and frequencies (percentages)) was used to represent the data. The chi-squared test was used to determine any significant differences in the distribution of the students’ characteristics between the knowledge levels. A knowledge score was calculated from nine true or false questions about genetics and PGX. Three knowledge levels were created based on the number of correct answers: good (7–9 correct answers), fair (4–6 correct answers) and poor (3 or less correct answers). For the attitudes, a 5-point Likert scale of strongly agree, agree, strongly disagree, disagree and neutral was collapsed into agree, disagree and neutral for ease of analysis and interpretation. The frequency distribution of the Likert scale results was reported as percentages to recognize the challenging areas of genomic medicine and PGX that students identify with.

This study had been approved by the Social Science Research Ethics Committee of United Arab Emirates University (UAEU) ERS_2017_5671. Participants were asked to read the information sheet of the study as well as to sign the consent form before starting the survey.

## 3. Results

### 3.1. Students’ Demographic and Academic Characteristics

A total of 510 students consented and completed the questionnaire between December 2018 and October 2019. Of the participating students, 82.7% were female. The mean (SD) age was 22 (±4.7) years old, and 76.1% were between the ages of 18 and 28. Most responses (68.6%) came from students who were studying in universities located in Al Ain city. Of the students, 52.2% were studying medicine and 29.3% were studying pharmacy. Most of the students (73.9%) were in pursuit of a bachelor’s degree and were in third and fourth year (22.2 and 23.4%, respectively). Table 1 summarizes the students’ demographic and academic characteristics.

### 3.2. Assessment of General Knowledge of Students on Genomic Medicine and PGX

Only 4.2% responded correctly to all the knowledge questions. The highest proportion of correct answers was for question 6 about the impact of genetics’ impact on drug response, and the lowest proportion of correct answers was for question 8 regarding cell composition. Table 2 summarizes the results of the knowledge questions.

Table 3 summarizes the distribution of the knowledge score and levels by the demographic and academic characteristics of the students. The mean knowledge score (SD) for all students was 5.4 (±2.7). The mean knowledge scores for students studying medicine and pharmacy were 5.5 (±2.7) and 5.6 (±2.7), respectively. The mean score of students in pursuit of a bachelor’s was 6.4 (±1.7), master’s: 5.9 (±1.5) and PhD: 6.6 (±1.2). A higher mean knowledge score was found in students who completed a PGX or pharmacogenetics related training or education (6.5; ±2.2) than those who did not (5.6; ±2.1).

There were significant differences in the levels of knowledge by the year of study of bachelor’s degree students, the completion status of training or education in PGX or pharmacogenetics and the completion of an internship or study abroad program (*p*-values < 0.05). Higher proportions of bachelor’s students in years 2–6 reported good to fair levels of knowledge. Higher proportions of master’s students in years 1 and 2 reported fair levels of knowledge. Of the students who completed PGX/pharmacogenetics training or education, 62.5% reported good level of knowledge. Out of the 510 students, 406 (79.6%) reported to have completed an internship or study abroad program. Totals of 47% and 41.6% of these students reported good and fair levels of knowledge, respectively.

### 3.3. Attitudes towards Genomic Medicine and PGX

Results on the attitudes towards genomic medicine and PGX were categorized into five categories: views and considerations, desire to participate, accessibility and availability of genetic tests, concerns and ethics, and, lastly, outlooks on the future.

#### 3.3.1. Views and Considerations on Genomic Medicine and PGX

The majority of students (82.7%) would consider having genetic testing done at some point in their life to find out their future risk of developing genetic diseases, whereas 74.7% would only like to know their susceptibility to diseases that have current interventions for protection. When asked if they prefer a pharmacist or physician to explain their genome report, 79.4% preferred a physician, while 44.8% preferred a pharmacist (Figure 1).

#### 3.3.2. Desire to Participate

A high percentage of students (78.1%) stated to be interested in participating in genetic research. Over three quarters of students (79.4%) indicated that they would be interested in attending a course or seminar for PGX education (Figure 2).

#### 3.3.3. Accessibility and Availability of Genetic Testing

The vast majority of students, respectively, 96.4% and 66.8%, reflected positive attitudes towards the availability and accessibility of genetic tests. However, 57.5% did agree that the availability of genetic tests could be problematic for insurance companies and future employers (Figure 3).

#### 3.3.4. Concerns and Ethics Regarding Genomic Medicine and PGX

The highest concern (66.8%) was that genomics could be exploited and used as means of discrimination (Figure 4). The next concern by percentage (40.2%) was due to issues of confidentiality, and a similar percentage (38.1%) were skeptical toward PGX testing due to a possibility of getting gene information unrelated to treatment.

#### 3.3.5. Outlooks on the Future of Genomic Medicine and PGX

The majority of students were optimistic about the future; 87.1% believing medicine will be more personalized. Most of them (89.9%) had a positive view on genetic testing and agreed that the government should invest more money into its development. Moreover, 73.2% thought more time should be dedicated towards studying PGX.

The top two barriers that students identified to the implementation of genomic medicine and PGX were lack of training or education (59.7%) and lack of clinical guidelines (58.7%). The next two highly perceived barriers were cost of testing and lack of testing services (46.3% and 44.7%, respectively). Other answers included lack of awareness and cultural/religious inhibitions. In order to improve future education on genomic medicine and PGX, students were asked for their preferred method of learning. The majority of students, 70.8%, preferred workshops or seminars while 34.2% and 30% preferred internet based learning and self-directed learning, respectively. Others preferred learning during their internship year (37.6%).

## 4. Discussion

The majority of medical and health science students in the UAE had a positive attitudes toward genomic medicine and PGX; they would consider having genetic testing done at some point in their life to find out their future risk of developing genetic diseases. Nevertheless, they had a fair level of knowledge about genomic medicine and PGX.

Dearth of knowledge on genomic medicine and PGX is one of the identified barriers and challenges for the full implementation of genomic medicine and PGX. Studies denoted that healthcare providers had a gap in their knowledge about genomic medicine and PGX [7,20,21]. Medical and health science students are the future adopters of genomic medicine and pharmacogenomics. Therefore, it is crucial to identify the students’ knowledge and attitudes toward genomic medicine and PGX in an early stage so policy makers can intervene and strategize the roadmap for the full implementation of genomic medicine and PGX in the UAE.

Most of the student in our sample did not demonstrate a good level of knowledge in the area of genomic medicine and pharmacogenomics, which could reflect a gap in the educational landscape of genomic medicine and pharmacogenomics in the UAE. This identified gap is aligned with what other investigators had identified in undergraduate medical students in southeast Europe and the United Kingdom [2,9].

We found significant statistical differences between the level of knowledge of the undergraduates on genomic medicine by the year of study, and this can partly be attributed to the fact that, based on mapping of the curriculum offered by the colleges and departments in the UAE, genetic and PGX courses available to the medical and health science students in the curriculum are incorporated in the curriculum starting from second year. This mimics the trend of genetics and PGX education in the US and Canadian medical schools [22]. Additional significant differences were found in our study between the level of knowledge and engagement in a training or educational activity pertaining to genomic medicine or PGX and with the completion of an internship or study program abroad. This finding underpins the infancy of the universities’ omics programs in the UAE and articulates the urgency in revisiting these programs to avoid the bottleneck situation warned against by the International Society of Pharmacogenomics in their recommendations to the deans of medical and health sciences schools [12].

We anchored a positive prospect in terms of the principles of PGX in our cohort; around 90% of the students articulated that genetic changes affect responses to drugs. This aligns with the positive outcome reported by Talwar, D et al. in their systematic literature review [23]. By the same token, students in our sample and the students of pharmacy in Jordan and West Bank of Palestine lagged behind in denoting the pharmacogenomics’ recommendation of the FDA [24].

Medical and health science students in the UAE are united in terms of their attitudes toward genetic tests under the same banner with medical and health science students worldwide. In our sample, the majority of the students (82.7%) would consider having genetic testing done at some point in their life to find out their future risk of developing genetic diseases. In a study conducted by Laskey S. et al. among African American and other marginal students, 95% of them endorsed genetic testing for preventive care [25]. Interrelating attitudes were found among college students in the Kingdom of Saudi Arabia and Greece [26,27]. Nevertheless, 74.7% of the students in our sample would only like to know their susceptibility to diseases that have current interventions for protection and that synchronized with the Common-Sense Model of Self-Regulation (CSM) framework for understanding illness self-management, in which students can formulate action plans in response to the threat of genetic tests’ results [28,29].

The overwhelming majority of the students in our sample (around 80%) selected the physician to fill the role of explaining the report of the genetic tests to them, while around 45% of them voted for the pharmacist. This can be a stereotype of the current health system that the students had trained in as well as a reflection of their limited knowledge. Researches proposed a partnership between pharmacists, physicians, and genetic counselors as a model to adjust for the gap in knowledge [30,31].

Students in our sample stated a myriad of legal and ethical concerns and liabilities. They voiced concerns that the availability of genetic tests could be problematic for insurance companies and future employers. These concerns match those of students in the USA, KSA, Qatar and Greece [25,26,27,32]. A heuristic qualitative study conducted in Belgium, explored the direct and indirect worries of genetic tests and concluded that legislative powers need to be clear and subtle to relieve these concerns about genetic discrimination [33].

The majority of students in our sample were optimistic about the future and believed that medicine in the UAE will be more personalized. Most of them agreed that the government should invest more money into its implementation and more time should be dedicated towards tutoring PGX. These stands boost the sporadic effort to implement personalized medicine in the UAE in particular and the GCC and Middle East and North Africa (MENA) region in a wider spectrum. A study by Shah, S. N. and Shaheen, S foresees the UAE as a fruitful landscape in the genomic era as the UAE is a host to a substantial expat population which translates to versatility in phenotypes in addition to the UAE locals and their unique signature genetic traits [34]. Another study by Mitropoulos, K. et al. shed light on success stories on the implementation of genomic medicine, and, in their article, they recounted PGX research that launched in 1996 in the UAE and led to the discovery of many novel variants [35].

Students in the UAE are eager for literacy in genomic medicine and PGX, and they highlighted workshops, seminars and internship to be their preferred pedagogy. The students ranked internet-based courses as their third preference in educational approach, which can craft the strategy to remedy the current gap in knowledge. Existing resources on the Internet consolidate this reciprocity of knowledge [1,36,37,38,39,40,41]. Moreover, we explored the students’ perceived barriers to the full implementation of genomic medicine and PGX in the UAE. Students in our sample ranked lack of training and education as the first barrier. The breadth of research tackled this barrier. Richard t. et al. highlighted in their paper the robust role of PGX education as a panacea toward generating well-informed clinicians who will champion personalized medicine [42]. The students also foresee lack of clinical guidelines, cost of testing, lack of infrastructure as well as lack of community awareness as a bundle of barriers deterring the full implementation of genomic medicine and PGX in the UAE. Corresponding research studies tackled the same barriers and investigated strategies towards overcoming these barriers [43,44].

Assessing the attitudes and knowledge of medical and health science students in the UAE about genomic medicine and PGX is an added tool to the implementation kit needed to construct a roadmap for the full implementation of genomic medicine and pharmacogenomics in the UAE. It empowers stakeholders to tackle the gaps in knowledge and conquer the barriers and challenges.

The inherited bias of information bias and selection bias will be a limitation that had been accepted by previous studies. Snowball sampling is prone to selection bias or community bias, unknown sampling population size, and hence difficulty in calculating an accurate response rate. To address these limitations, we scanned all the medical and health science universities in the UAE and employed random selection sampling techniques. However, we had scarce representation of the Northern Emirates, and this might impact the generalizability of our findings.

## 5. Conclusions

The periodic assessment of the knowledge and attitudes of students of medical and health sciences schools in the UAE captures the gaps and harnesses measures to address these gaps. Stakeholders in the UAE must strive to acquaint their students with up-to-date knowledge of genomic medicine and PGX. Students of today are the champions of personalized medicine tomorrow. We recommend updating the curriculum of the medical and health sciences under the supervision of the experts in the field and in line with accreditation bodies. We propose stand-alone courses in genomic medicine and pharmacogenomics for both under- and post-graduate medical and health science students. We recommend initiating a rapport between academia and health setting to impute knowledge and translate knowledge into practice.

## Figures and Tables

**Figure 1 jpm-10-00191-f001:**
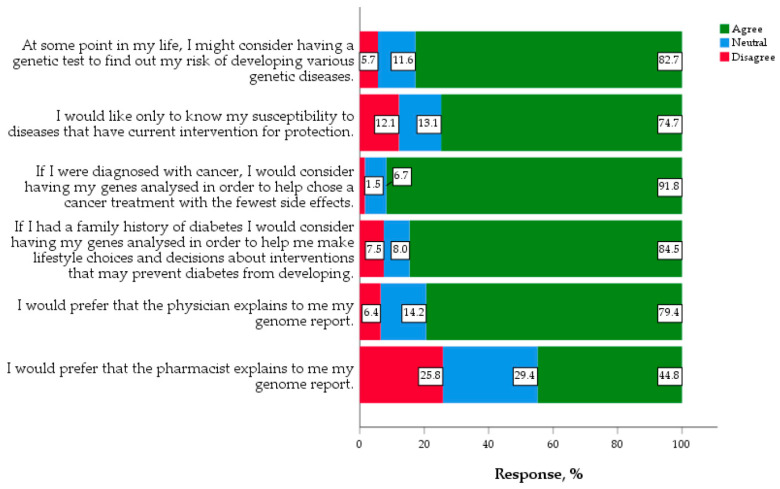
Views and considerations on genomic medicine and PGX (*n* = 388).

**Figure 2 jpm-10-00191-f002:**
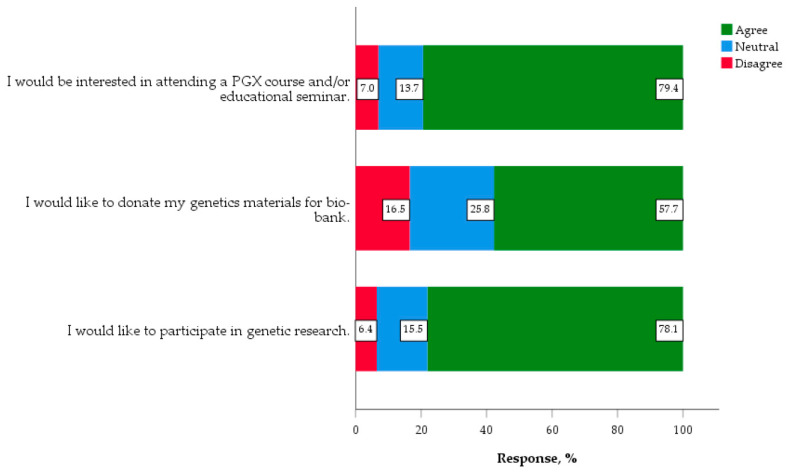
Desire to participate in genetic and PGX research (*n* = 388).

**Figure 3 jpm-10-00191-f003:**
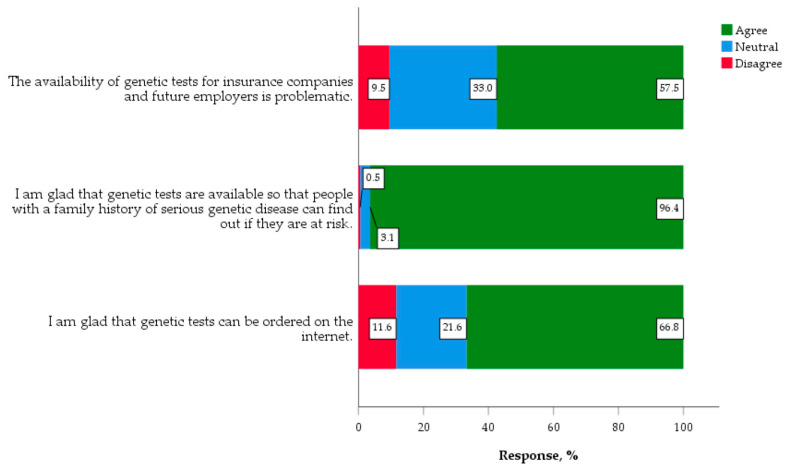
Accessibility and availability of genetic testing (*n* = 388).

**Figure 4 jpm-10-00191-f004:**
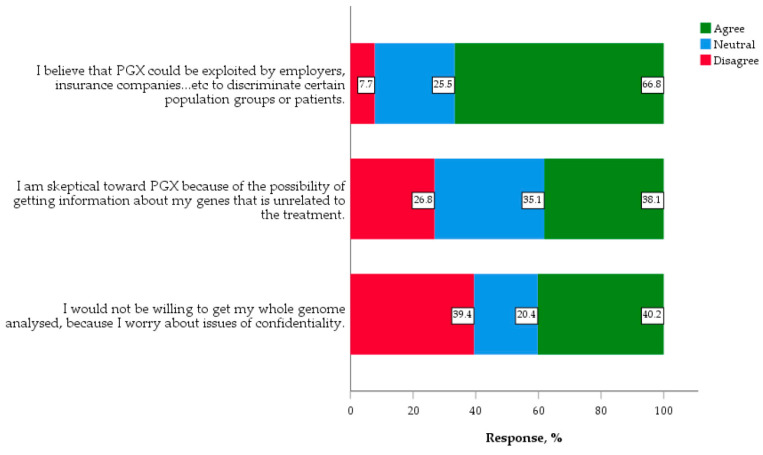
Concerns and ethics regarding genomic medicine and PGX (*n* = 388).

**Table 1 jpm-10-00191-t001:** Students’ demographic and academic characteristics (*n* = 510).

	Count (%)
Gender	
*Female*	421 (82.7%)
*Male*	88 (17.3%)
Age Group	
*<18*	69 (13.5%)
*18–28*	388 (76.1%)
*29–39*	38 (7.5%)
*40–50*	6 (1.2%)
University Location	
*Al Ain*	245 (68.6%)
*Dubai*	83 (16.5%)
*Sharjah*	55 (10.9%)
*Abu Dhabi*	17 (3.4%)
*Ajman*	1 (0.2%)
*Fujairah*	1 (0.2%)
*Ras Al Khaimah*	1 (0.2%)
Program	
*Medicine*	265 (52.2%)
*Pharmacy*	149 (29.3%)
*Laboratory*	35 (6.9%)
*Other ^a^*	59 (11.6%)

*^a^* medical imaging, radiology, radiography, biochemistry, biomedical sciences, dentistry, pharmacology, physiology, psychology, public health, occupational health.

**Table 2 jpm-10-00191-t002:** Questions assessing genomic medicine and pharmacogenomics’ knowledge among students (*n* = 506).

Knowledge Questions	Correct Answer	Answered “True” *n* (%)	Answered “False” *n* (%)	Answered “Do not Know” *n* (%)
1. Humans have 48 chromosomes.	False	149	350	7
		(29.4%)	(69.2%)	(1.4%)
2. Adenine (A) only pairs with cytosine (C) and Thymine (T) only pairs with Guanine (G).	False	79	385	42
		(15.6%)	(76.1%)	(8.3%)
3. Pharmacogenomics seeks to individualize therapy based on patient’s genetic profile.	True	418	12	76
		(82.6%)	(2.4%)	(15%)
4. Genetic changes can cause adverse reactions.	True	426	20	60
		(84.2%)	(4.0%)	(11.9%)
5. Pharmacogenomics testing is recommended by FDA for certain drugs.	True	261	25	220
		(51.6%)	(4.9%)	(43.5%)
6. Genetic changes can affect the patient’s response to certain drug.	True	455	11	40
		(89.9%)	(2.2%)	(7.9%)
7. Genes can be activated or deactivated by other genes.	True	412	16	78
		(81.4%)	(3.2%)	(15.4%)
8. Every cell of the body contains the whole genome.	False	314	112	80
		(62.1%)	(22.1%)	(15.8%)
9. Environmental factors, such as cigarette smoke, can affect gene activity.	True	423	43	40
		(83.6%)	(8.5%)	(7.9%)

**Table 3 jpm-10-00191-t003:** Comparison of the level of knowledge by demographic and academic characteristics.

		Level of Knowledge			
	Mean Score (± SD)	Good	Fair	Poor	*p*-Value
Overall	5.4 (± 2.7)	219 (42.9%)	191 (37.5%)	100 (19.6%)	
Gender					0.47
*Female*	5.3 (± 2.7)	176 (41.8%)	161 (38.2%)	84 (20.0%)	
*Male*	5.6 (± 2.6)	43 (48.9%)	30 (34.1%)	15 (17.0%)	
Age group					0.56
*<18*	5.0 (± 2.5)	23 (33.3%)	31 (44.9%)	15 (21.7%)	
*18–28*	5.5 (± 2.8)	177 (45.6%)	137 (35.3%)	74 (19.1%)	
*29–39*	5.3 (± 2.8)	14 (36.8%)	16 (42.1%)	8 (21.1%)	
*40–50*	5.0 (± 2.6)	2 (33.3%)	3 (50.0%)	1 (16.7%)	
Program					0.12
*Medicine*	5.5 (± 2.7)	123 (46.4%)	95 (35.8%)	47 (17.7%)	
*Pharmacy*	5.6 (± 2.7)	69 (46.3%)	52 (34.9%)	28 (18.8%)	
*Laboratory*	4.7 (± 2.9)	11 (31.4%)	15 (42.9%)	9 (25.7%)	
*Other*	4.8 (± 2.4)	16 (27.1%)	29 (49.2%)	14 (23.7%)	
Degree					0.44
*Bachelor*	6.4 (± 1.7)	185 (44.5%)	148 (35.6%)	83 (20.0%)	
*Master*	5.9 (± 1.5)	14 (33.3%)	20 (47.6%)	8 (19.0%)	
*PhD*	6.6 (± 1.2)	19 (40.4%)	20 (42.6%)	8 (17.0%)	
*Other*	5.8 (± 1.0)	1 (25.0%)	3 (75.0%)	0 (0.0%)	
Year of study (Bachelor)					0.00 *
*First*	5.1 (± 2.0)	11 (5.9%)	29 (19.6%)	13 (15.7%)	
*Second*	6.4 (± 1.7)	37 (20.0%)	22 (14.9%)	15 (18.1%)	
*Third*	7.0 (± 1.4)	52 (28.1%)	26 (17.6%)	22 (26.5%)	
*Fourth*	6.6 (± 1.6)	55 (29.7%)	35 (26.3%)	20 (24.1%)	
*Fifth*	6.5 (± 1.5)	23 (12.4%)	12 (8.1%)	7 (8.4%)	
*Sixth*	6.1 (± 1.3)	5 (2.7%)	12 (8.1%)	3 (3.6%)	
*Other*	5.8 (± 1.2)	2 (1.1%)	12 (8.1%)	3 (3.6%)	
Year of study (Master)					0.35
*First*	5.8 (± 1.4)	5 (35.7%)	9 (45.0%)	3 (37.5%)	
*Second*	5.8 (± 1.6)	5 (35.7%)	10 (50.0%)	3 (37.5%)	
*Third*	6.3 (± 1.2)	2 (14.3%)	1 (5.0%)	2 (25.0%)	
*Other*	7.5 (± 0.7)	2 (14.3%)	0 (0.0%)	0 (0.0%)	
Year of study (PhD)					0.08
*First*	6.2 (± 1.1)	7 (36.8%)	12 (60.0%)	1 (12.5%)	
*Second*	6.9 (± 1.2)	5 (26.3%)	4 (20.0%)	1 (12.5%)	
*Third*	7.0 (± 1.3)	3 (15.8%)	3 (15.0%)	2 (25.0%)	
*Fourth*	7.5 (± 0.6)	4 (21.1%)	0 (0.0%)	4 (50.0%)	
*Fifth*	5.0 (± 0.0)	0 (0.0%)	1 (5.0%)	0 (0.0%)	
Previous exposure to genetic issues					0.56
*Yes*	5.9 (± 2.1)	94 (45.2%)	92 (44.2%)	22 (10.6%)	
*No*	6.0 (± 2.2)	125 (49.6%)	99 (39.3%)	28 (11.1%)	
Completed PGX/pharmacogenetics training or education					0.00 *
*Yes*	6.5 (± 2.2)	110 (62.5%)	51 (29.0%)	15 (8.5%)	
*No*	5.6 (± 2.1)	109 (38.4%)	140 (49.3%)	35 (12.3%)	
Completed internship or study abroad program					0.00 *
*Yes*	5.9 (± 2.2)	191 (47.0%)	169 (41.6%)	46 (11.3%)	
*No*	3.2 (± 3.4)	29 (27.4%)	22 (20.8%)	55 (51.9%)	

* significant *p*-value < 0.05.

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
