# Peer review of "Knowledge and Attitudes of Medical and Health Science Students in the United Arab Emirates toward Genomic Medicine and Pharmacogenomics: A Cross-Sectional Study"

_jpm, 2020, doi:10.3390/jpm10040191_

Round 1
Reviewer 1 Report
Pharmacogenetics is a rapidly growing field studying of how genetic differences influence the variability of individual patient responses to drugs, aims to distinguish responders from non-responders and predict those in whom toxicity and It can be regarded as the 21st century's answer for the rational use of drugs - the right drug to the right patient at right dose. The most important barriers delaying clinical uptake and application of pharmacogenomics is lack of knowledge and insufficient education of health professionals regarding pharmacogenetics and genomics rather than technical issue. Since the integration of pharmacogenetics (PGx) testing into routine care will, in part, depend upon the physicians’, pharmacists’ and as well as patients’ acceptance the education of scientists, healthcare professionals and publics in genetics is crucial for appropriate application of pharmacogenetics to integrate into healthcare system. . Understanding of implication of personalized medicine and so pharmacogenetics in drug reactions incidence will allow healthcare professionals to be an integral part of the new age of personalized medicine. Rahma and his colleague’s by means of cross-sectional survey study aimed to explore how well medical and health sciences students in UAE perceive importance and application of genomic medicine and pharmacogenetics. To this end, this manuscript highlights the potential opportunities and applications of PGx in clinical practice. Over all the authors do a nice job! The manuscript is very well written and the introduction provides a good, generalized background about the importance of PGx
Author Response
17th October 2020
Manuscript ID: jpm-957448
Knowledge and Attitude of Medical and Health Sciences Students in United Arab Emirates toward Genomic Medicine and Pharmacogenomics: a Cross-sectional Study” We would like to thank the editor and the reviewer for assessing our work and for their valuable feedback and suggestions. Indeed, addressing the raised comments and suggestions have improved the quality and readability of our manuscript. We have provided a point-by-point response to the request and comments raised by the editorial team and reviewer.
Reply to the Reviewers' Comments to Author
Reviewer 1:
Rahma and his colleague’s by means of cross-sectional survey study aimed to explore how well medical and health sciences students in UAE perceive importance and application of genomic medicine and pharmacogenetics. To this end, this manuscript highlights the potential opportunities and applications of PGx in clinical practice. Over all the authors do a nice job! The manuscript is very well written, and the introduction provides a good, generalized background about the importance of PGx
Thank you very much for your valuable comments.
Reviewer 2 Report
Dear Authors,
Thank you for your contribution to the field of pharmacogenomics. I enjoyed reading your paper and have a few comments that you could consider for improvement.
Minor Criticisms
- Missing exponent sign in formula (1.96 2) [line 102, should be 1.96^2]
- 48% prevalence of what? [line 103]
- In the methods, it would be helpful if the authors would expand on exactly what analyses were conducted on the Likert data.
- Section 2.3.4 of the results: the authors should consider expanding this to include more specific results rather than just saying that "all concerns..."
- The authors mention bias in the limitations, but I felt that it would be of benefit to include a deeper discussion of the bias from snowball sampling.
Author Response
17th October 2020
Manuscript ID: jpm-957448
Knowledge and Attitude of Medical and Health Sciences Students in United Arab Emirates toward Genomic Medicine and Pharmacogenomics: a Cross-sectional Study” We would like to thank the editor and the reviewer for assessing our work and for their valuable feedback and suggestions. Indeed, addressing the raised comments and suggestions have improved the quality and readability of our manuscript. We have provided a point-by-point response to the request and comments raised by the editorial team and reviewer.
Reply to the Reviewers' Comments to Author
Reviewer 2:
Thank you for your contribution to the field of pharmacogenomics. I enjoyed reading your paper and have a few comments that you could consider for improvement.
Thank you very much for your supporting comments.
Minor Criticisms:
- Missing exponent sign in formula (1.96 2) [line 102, should be 1.96^2]
Thank you so much for noticing that and accept my apology for missing this point. Sign Superscript (Line 102) 1.962 x P (1-P) / d2), I also fixed and superscript the (d2 ).
- 48% prevalence of what? [line 103]
My apology, I missed out the details, the prevalence of the knowledge of genomics among medical and health sciences students now is inserted in the text. (line 103)
This is the table of the literature review for the level of knowledge of genomics among students.
|
|
Citation |
Prevalence |
Response rate |
Comments |
|
1
|
Merdad L, Aldakhil L, Gadi R, Assidi M, Saddick SY, Abuzenadah A, Vaught J, Buhmeida A, Al-Qahtani MH. Assessment of knowledge about biobanking among healthcare students and their willingness to donate biospecimens. BMC medical ethics. 2017 Dec;18(1):32. |
44 % (Genome project)
|
86% |
KSA / STUDENTS
(knowledge) |
|
2
|
Read CY, Ward LD. Misconceptions About Genomics Among Nursing Faculty and Students. Nurse educator. 2018 Jul 1;43(4):196-200. |
42% for students |
----- |
Nursing Faculty/Students |
|
3
|
McGruder C. Evaluation of current knowledge of genetics among dental students, residents and dental hygiene students. |
70 % |
81% |
USA / dental students, residents and dental hygiene students. |
|
4
|
Moen M, Lamba J. Assessment of healthcare students’ views on pharmacogenomics at the University of Minnesota. Pharmacogenomics. 2012 Oct;13(13):1537-45. |
11.1% |
------ |
USA pharmacy, nursing and medical students |
|
5
|
Zhou S. Teaching of clinical pharmacogenetics for pharmacy studentsat the National University of Singapore. Pharmacy Education. 2005;5. |
70.8% |
|
Singapore
Pharmacy students |
|
|
Average (Students)
|
48% |
84% |
|
- In the methods, it would be helpful if the authors would expand on exactly what analyses were conducted on the Likert data.
We used IBM Statistical Package for the Social Sciences (SPSS) Statistics version 26 to analyze the data. Descriptive statistics (means (standard deviation, SD) and frequencies (percentages)) was used to represent the data. (line 107-109)
For the attitudes, a 5-point Likert scale of strongly agree, agree, strongly disagree, disagree and neutral was collapsed into agree, disagree and neutral for ease of analysis and interpretation. The frequency distribution of the Likert scale results was reported as percentages to recognize the challenging areas of genomic medicine and PGX that students identify with. (line 113-117)
Section 2.3.4 of the results: the authors should consider expanding this to include more specific results rather than just saying that "all concerns..."
Thank you very much for spotting this, based on your comments we deleted “All concerns were met with higher percentages of agreement than disagreement” (line 169)
We replaced it with “The highest concern (66.8%) was that genomics could be exploited and used as means of discrimination (Figure 4). The next concern by percentage (40.2%) was due to issues of confidentiality and a similar percentage of 38.1% were skeptical toward PGX testing due to a possibility of getting gene information unrelated to the treatment.” (line 169-173)
- The authors mention bias in the limitations, but I felt that it would be of benefit to include a deeper discussion of the bias from snowball sampling.
Thank you so much, this comment added clarity and honesty to the limitation. We added the following:
“Snowball sampling is prone to selection bias or community bias, unknown sampling population size and hence difficulty in calculating accurate response rate. To address these limitations, we scanned all the medical and health science universities in UAE and employed random selection sampling techniques.” (line 273-276)